# Mixing indistinguishable systems leads to a quantum Gibbs paradox

Benjamin Yadin [1,2 ✉], Benjamin Morris [1 ✉] & Gerardo Adesso [1 ✉]

The classical Gibbs paradox concerns the entropy change upon mixing two gases. Whether an observer assigns an entropy increase to the process depends on their ability to distinguish the gases. A resolution is that an "ignorant" observer, who cannot distinguish the gases, has no way of extracting work by mixing them. Moving the thought experiment into the quantum realm, we reveal new and surprising behaviour: the ignorant observer can extract work from mixing different gases, even if the gases cannot be directly distinguished. Moreover, in the macroscopic limit, the quantum case diverges from the classical ideal gas: as much work can be extracted as if the gases were fully distinguishable. We show that the ignorant observer assigns more microstates to the system than found by naive counting in semiclassical statistical mechanics. This demonstrates the importance of accounting for the level of knowledge of an observer, and its implications for genuinely quantum modifications to thermodynamics.

[1] School of Mathematical Sciences and Centre for the Mathematics and Theoretical Physics of Quantum Non-Equilibrium Systems, University Park, University of Nottingham, Nottingham, UK. [2] Wolfson College, University of Oxford, Oxford, UK. ✉email: benjamin.yadin@physics.ox.ac.uk; benjamin.morris@nottingham.ac.uk; gerardo.adesso@nottingham.ac.uk

D espite its phenomenological beginnings, thermodynamics has been inextricably linked throughout the past century with the abstract concept of information. Such connections have proven essential for solving paradoxes in a variety of thought experiments, notably including Maxwell's demon[1] and Loschmidt's paradox[2]. This integration between classical thermodynamics and information is also one of the main motivating factors in extending the theory to the quantum realm, where information held by the observer plays a similarly fundamental role[3].

This work is concerned with the transition from classical to quantum thermodynamics in the context of the Gibbs paradox[4–6]. This thought experiment considers two gases on either side of a box, separated by a partition and with equal volume and pressure on each side. If the gases are identical, then the box is already in thermal equilibrium, and nothing changes after removal of the partition. If the gases are distinct, then they mix and expand to fill the volume independently, approaching thermal equilibrium with a corresponding entropy increase. The (supposed) paradox can be summarised as follows: what if the gases differ in some unobservable or negligible way—should we ascribe an entropy increase to the mixing process or not? This question sits uncomfortably with the view that thermodynamical entropy is an objective physical quantity.

Various resolutions have been described, from phenomenological thermodynamics to statistical mechanics perspectives, and continue to be analysed[6–8]. A crucial insight by Jaynes[9] assuages our discomfort at the observer-dependent nature of the entropy change. For an informed observer, who sees the difference between the gases, the entropy increase has physical significance in terms of the work extractable through the mixing process—in principle, they can build a device that couples to the two gases separately (for example, through a semi-permeable membrane) and thus let each gas do work on an external weight independently. An ignorant observer, who has no access to the distinguishing degree of freedom, has no device in their laboratory that can exploit the difference between the gases, and so cannot extract work. For Jaynes, there is no paradox as long as one considers the abilities of the experimenter—a viewpoint central to the present work.

A study of Gibbs mixing for identical quantum bosons or fermions is motivated by recognising that the laws of thermodynamics must be modified to account for quantum effects such as coherence[10], which can lead to enhanced performance of thermal machines[11–13]. The thermodynamical implications of identical quantum particles have received renewed interest for applications such as Szilard engines[14,15], thermodynamical cycles[16,17] and energy transfer from boson bunching[18]. Moreover, the particular quantum properties of identical particles, including entanglement, can be valuable resources in quantum information processing tasks[19–21].

In this work, we consider a toy model of an ideal gas with non-interacting quantum particles, distinguishing the two gases by a spin-like degree of freedom. We describe the mixing processes that can be performed by both informed and ignorant observers, taking into account their different levels of control, from which we can calculate the corresponding entropy changes and thus work extractable by each observer. For the informed observer, we recover the same results as obtained by classical statistical mechanics arguments. However, for the ignorant observer, there is a marked divergence from the classical case. Counter-intuitively, the ignorant observer can typically extract more work from distinguishable gases—even though they appear indistinguishable—than from truly identical gases. In the continuum and large particle number limit which classically recovers the ideal gas, this divergence is maximal: the ignorant observer can extract as much

work from apparently indistinguishable gases as the informed observer. Our analysis hinges on the symmetry properties of quantum states under permutations of particles. For the ignorant observer, these properties lead to non-trivial restrictions on the possible work extraction processes. Viewed another way, the microstates of the system described by the ignorant observer are highly non-classical entangled states. This implies a fundamentally different way of counting microstates, and therefore computing entropies, from what is done classically or even in semi-classical treatments of quantum gases. Therefore we uncover a genuinely quantum thermodynamical effect in the Gibbs mixing scenario.

## Results

**Set-up**. We consider a gas of $N$ particles inside a box, such that each particle has a position degree of freedom, denoted $x$, and a second degree of freedom which distinguishes the gases. Since we only consider the case of two types of gases, this is a two-dimensional degree of freedom and we refer to this as the 'spin' $s$ (although it need not be an actual angular momentum). Classically, the two spin labels are $\uparrow$, $\downarrow$, and their quantum analogues are orthogonal states $|\uparrow\rangle, |\downarrow\rangle$.

Following the traditional presentation of the Gibbs paradox, the protocol starts with two independent gases on different sides of a box: $n$ on the left and $m = N − n$ on the right (see Fig. 1). Each side is initially thermalised with an external heat bath $B$ at temperature $T$.

In our toy model, each side of the box consists of $d/2$ 'cells' ($d$ is even) representing different states that can be occupied by each particle. These states are degenerate in energy, such that the Hamiltonian of the particles vanishes. This might seem like an

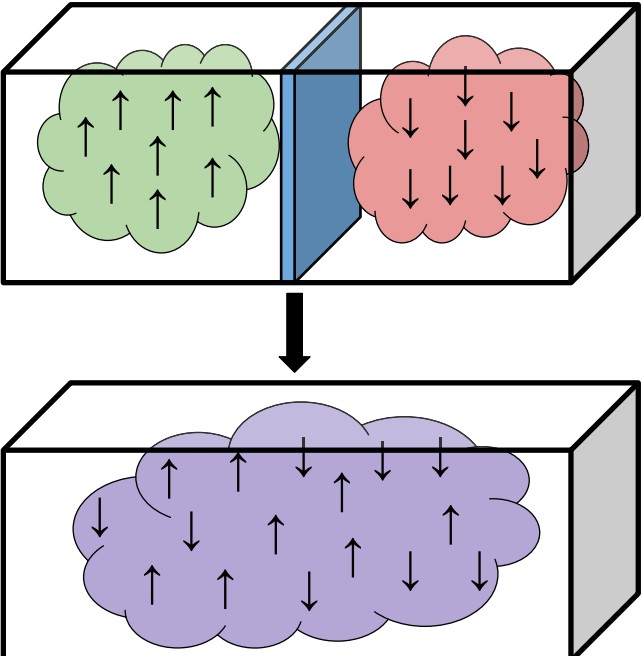

**Fig. 1 The Gibbs paradox.** Two distinct gases of $n$ particles at the same temperature and pressure are separated by a partition. This partition is removed and the gases are allowed to mix and reach equilibrium. Two observers calculating the entropy increase during the process disagree depending on their ability to distinguish the particles. An informed observer, who can measure the difference between the gases, calculates $2n \ln 2$, while an observer ignorant of the difference records no entropy change. In this work, we ask how the situation changes when classical particles are replaced by identical quantum particles.

**Table 1 Summary of the observers' abilities.**

| Observer | Can | Cannot |
|---|---|---|
| Informed | Access the spin and spatial degrees of freedom | Change the number of up or down spins |
| Ignorant | Access the spatial degree of freedom | Access the spin degree of freedom |

unrealistic assumption; however, this model contains the purely combinatorial (or 'state-counting') statistical effects, first analysed by Boltzmann[22], that are known to recover the entropy changes for a classical ideal gas[8,23,24] using the principle of equal a priori probabilities. One could instead think of this setting as approximating a non-zero Hamiltonian in the high-temperature limit, such that each cell is equally likely to be occupied in a thermal state. Since the particle number is strictly fixed, we are working in the canonical ensemble (rather than the grand canonical ensemble).

Work extraction can be modelled in various ways in quantum thermodynamics. In the resource-theoretic approach based on thermal operations[25,26], one keeps track of all resources by treating the system (here, the particles), heat bath and work reservoir (or battery) as interacting quantum systems. The work reservoir is an additional system with non-degenerate Hamiltonian whose energy changes are associated with work done by or on the system (generalising the classical idea of a weight being lifted and lowered).

The gases on either side of the box start in a state of local equilibrium and via mixing approach global equilibrium. We therefore consider the extractable work to be given by the difference in non-equilibrium free energy $F$[27] between initial and final states, where $F(\rho) = \langle E \rangle_\rho - k_B T S(\rho)$, $\langle E \rangle_\rho = \text{tr}(\rho H)$ being the mean energy (zero in our case) and $S(\rho) = -\text{tr}(\rho \ln \rho)$ the von Neumann entropy in natural units. The extractable work in a process that takes $\rho$ to $\rho'$ is then

$$W \leq F(\rho) - F(\rho') = k_B T [S(\rho') - S(\rho)]. \quad (1)$$

In a classical reversible process, the extractable work is equal to the change in free energies. This is generally an over-simplification for small systems, in which work can be defined in various ways[28]—e.g. required to be deterministic in the resource theory context[25] or as a fluctuating random variable[29,30], requiring consideration of other varieties of free energy. However, Eq. (1) will turn out to be sufficient for our purposes in the sense of mean extractable work. We find the inequality to be saturable using thermal operations and characterise fluctuations around the mean in the latter part of our results section.

Our analysis compares the work extracted by two observers with different levels of knowledge: the informed observer, who can tell the difference between the two gases, and the ignorant observer, who cannot. The difference between these observers is that the former has access to the spin degree of freedom $s$, whereas the latter does not (summarised in Table 1).

It is important to point out that, for the informed observer, the spin acts as a 'passive' degree of freedom, meaning that it can be measured but not actively changed. In other words, the two types of gases cannot be converted into each other. This assumption is always implicitly present in discussions of the Gibbs paradox—without it, the distinguishing degree of freedom would constitute another subsystem with its own entropy changes. One could also describe the spin as an information-bearing degree of freedom[31]. The question is whether the information encoded within the spin state has an impact upon the thermodynamics of mixing.

**Classical case.** Classically, the microstates described by the informed observer are specified by counting how many particles

exist with each position $x$ and spin $s$—since the particles are indistinguishable[32]. The ignorant observer has a different state space given by coarse-graining these states—the classical equivalent of 'tracing out' the spin degree of freedom. Thus the ignorant observer can extract only as much work from two different gases as from a single gas, recovering Jaynes' original statement[9]. These intuitively obvious facts are shown by a formal construction of the state spaces in Supplementary Note 1. Paralleling our later quantum treatment, this establishes that the classical and quantum cases can be compared fairly.

The amount of extractable work in the classical case can be straightforwardly argued by state counting. Consider the gas initially on the left side—the number of ways of distributing $n$ particles among $d/2$ cells is $\binom{n + d/2 - 1}{n}$. In the thermal state, each configuration occurs with equal probability. Therefore the initial entropy, also including the gas on the right, is $\ln \binom{n + d/2 - 1}{n} + \ln \binom{m + d/2 - 1}{m}$. For distinguishable gases, each gas can deliver work independently, with an equal distribution over $\binom{n + d - 1}{n} \binom{m + d - 1}{m}$ configurations. For indistinguishable gases, the final thermal state is described as an equal distribution over all ways of putting $N = n + m$ particles into $d$ cells, of which there are $\binom{N + d - 1}{N}$. Hence the entropy change in each case is

$$\Delta S = \ln \binom{n + d - 1}{n} + \ln \binom{m + d - 1}{m} - \ln \binom{n + d/2 - 1}{n}$$
$$- \ln \binom{m + d/2 - 1}{m} \quad (\text{distinguishable}),$$
$$(2)$$

$$\Delta S = \ln \binom{N + d - 1}{N} - \ln \binom{n + d/2 - 1}{n}$$
$$- \ln \binom{m + d/2 - 1}{m} \quad (\text{indistinguishable}).$$
$$(3)$$

Note that $\Delta S \neq 0$ even in the indistinguishable case, which may seem at odds intuitively with the result for an ideal gas. However, one can check that $\Delta S = O(\ln N)$ in the limit of large $d$ (whereby the box becomes a continuum) and large $N$. This is negligible compared with the ideal gas expression of $N \ln 2$ for distinguishable gases[33] (See ref. [8], p. 43 for a more detailed discussion of this approximation). (Due to a subtle technicality with classical identical particles, formulas (2), (3) might be regarded as upper bounds to the true values—see Supplementary Note 1.) Note that a classical analogue of fermions can be made by importing the Pauli exclusion principle, so that two or more particles can never occupy the same cell. This has the effect of replacing the binomial coefficients of the form $\binom{N + d - 1}{N}$ in (2) and (3) by $\binom{d}{N}$.

**Quantum case.** Compared with the classical case, we must be more explicit about the role of the spin $s$ as a 'passive' degree of freedom for the informed observer. This observer may obtain information about the numbers of spin—↑ and spin—↓ particles.

Thus they can engineer spin-dependent operations conditional on these numbers, but cannot change the number of each spin.

For identical gases, the result is of course the same as for the ignorant observer, and the classical case (3). For distinguishable gases, each gas behaves as an independent subsystem; thus, the entropy changes are the same as for classical distinguishable gases (2).

The remainder of this section is devoted to the ignorant observer, for which we find a departure from the classical case.

The peculiarities of the quantum case stem from a careful look at the Hilbert space structure. The Hilbert space of a single particle is a product $\mathcal{H}_1 = \mathcal{H}_x \otimes \mathcal{H}_s$ of a part for the spatial degree of freedom $x$ and a part for the spin $s$. Since there are $d$ cell modes and two spin states, these parts have dimensions dim $\mathcal{H}_x = d$, dim $\mathcal{H}_s = 2$. For $N$ distinguishable particles, the state space would be $\mathcal{H}_1^{\otimes N}$. However, for bosons and fermions, which are quantum indistinguishable particles, states lie in the symmetric and antisymmetric subspaces, respectively (in first quantisation). This symmetry refers to the wavefunction under permutations of particles: for bosons, there is no change, whereas for fermions, each swap of a pair incurs a minus sign in the global phase. The physical Hilbert space of $N$ particles can then be written as

$$\mathcal{H}_N = P_{\pm}\left(\mathcal{H}_x^{\otimes N} \otimes \mathcal{H}_s^{\otimes N}\right), \tag{4}$$

where $P_{+(-)}$ is the projector onto the (anti-)symmetric subspace.

Since each particle carries a position and spin state, a permutation $\Pi$ of particles is applied simultaneously to these two parts: $\Pi$ acts on the above Hilbert space in the form $\Pi_x \otimes \Pi_s$. The requirement of an overall (anti-)symmetric wavefunction effectively couples these two degrees of freedom via their symmetries. For a familiar example, consider two particles. The spin state space can be broken down into the symmetric 'triplet' subspace spanned by $|\uparrow\uparrow\rangle$, $|\downarrow\downarrow\rangle$ and $|\uparrow\downarrow\rangle + |\downarrow\uparrow\rangle$, and the antisymmetric 'singlet' subspace consisting of $|\uparrow\downarrow\rangle - |\downarrow\uparrow\rangle$. For bosons, overall symmetry requires that a triplet spin state be paired with a symmetric spatial wavefunction, and a singlet spin state with an antisymmetric spatial function. For fermions, opposite symmetries are paired.

With more particles, the description is more complex, but the main idea of paired symmetries remains the same. Following ref. [34], our main tool is Schur-Weyl duality[35], which decomposes

$$\mathcal{H}_x^{\otimes N} = \bigoplus_\lambda \mathcal{H}_x^\lambda \otimes \mathcal{K}_x^\lambda, \tag{5}$$

where $\lambda$ runs over all Young diagrams of $N$ boxes and no more than $d$ rows (A Young diagram can be described simply by a non-increasing set of ($\leq d$) positive integers summing up to $N$). In technical terms, $\mathcal{H}_x^\lambda$ and $\mathcal{K}_x^\lambda$ carry irreducible representations of the unitary group $U(d)$ and the permutation group $S_N$ of $N$ particles, respectively. More concretely, a non-interacting unitary operation on the positions of all the particles, $u_x^{\otimes N}$, is represented in the decomposition (5) as an independent rotation within each of the $\mathcal{H}_x^\lambda$ spaces. The term 'irreducible' refers to the fact that each of these spaces may be fully explored by varying the unitary $u_x$. Similarly, a permutation of the particles in the spatial part of the wavefunction is represented by an action on each $\mathcal{K}_x^\lambda$ space. Thus each block labelled by $\lambda$ in the decomposition (5) has a specific type of permutation symmetry.

The same decomposition works for the spin part $\mathcal{H}_s^{\otimes N}$. However, since this degree of freedom is two-dimensional, each $\lambda$ is constrained to have no more than two rows. We can think of $s$ as describing a total angular momentum formed of $N$ spin-1/2

particles, and in fact $\lambda$ can be replaced by a total angular momentum eigenvalue $J$ varying over the range $N/2, N/2 - 1, \ldots$.

After putting the spatial and spin decompositions together, projecting onto the overall (anti-)symmetric subspace causes the symmetries of the two parts to be linked. For bosons, the $\lambda$ label for $x$ and $s$ must be the same; for fermions, they are transposes of each other (i.e. related by interchanging rows and columns). This results in the form

$$\mathcal{H}_N = \bigoplus_\lambda \mathcal{H}_x^\lambda \otimes \mathcal{H}_s^\lambda \quad \text{for bosons,} \tag{6}$$

$$\mathcal{H}_N = \bigoplus_\lambda \mathcal{H}_x^{\lambda^T} \otimes \mathcal{H}_s^\lambda \quad \text{for fermions.} \tag{7}$$

Instead of the label $\lambda$, from now on we use the angular momentum number $J$ and generally write this decomposition as $\bigoplus_J \mathcal{H}_x^J \otimes \mathcal{H}_s^J$—bearing in mind that $\mathcal{H}_x^J$ is different for bosons and fermions. In terms of the earlier $N = 2$ example, $J = 1$ corresponds to the spin triplet subspace, and $J = 0$ to the spin singlet.

Another way of describing the decomposition (6) is that it provides a convenient basis $|J, q\rangle_x |J, M\rangle_s |\phi_J\rangle_{xs}$, known as the Schur basis[36]. Here, $\{|J, q\rangle_x\}_q$ is a basis for $\mathcal{H}_x^J$ and $\{|J, M\rangle_s\}_M$ a basis for $\mathcal{H}_s^J$. $M = -J, -J + 1, \ldots, J$ can be interpreted as the total angular momentum quantum number along the z-axis. $|\phi_J\rangle_{xs} \in \mathcal{K}_x^J \otimes \mathcal{K}_s^J$ is a state shared between the $x$ and $s$ degrees of freedom.

We now consider how the state thermalises for the ignorant observer. Since the ignorant observer cannot interact with spin, their effective state space is described by tracing out the factor $\mathcal{H}_s$ for each particle. In terms of the decomposition (6) and corresponding basis described above, this means that an initial density matrix $\rho$, after tracing out $s$, is of the form

$$\rho_x := \text{tr}_s \rho = \bigoplus_J p_J \rho_x^J \otimes \text{tr}_s |\phi_J\rangle\langle\phi_J|_{xs}, \tag{8}$$

where $\rho_x^J$ is a density matrix on $\mathcal{H}_x^J$, occurring with probability $p_J$. Note that there is no coherence between different values of $J$, and that the components $\rho_x^J$ are mutually perfectly distinguishable by a measurement of their $J$.

Additionally, the allowed operations must preserve the bosonic or fermionic exchange symmetry. Any global unitary $U_{xBW}$, coupling the spatial degree of freedom of the particles to the heat bath and work reservoir, must therefore commute with permutations on the spatial part: $[U_{xBW}, \Pi_x] = 0$ for all $\Pi$. By Schur's Lemma, such a unitary decomposes as $U = \bigoplus_J U^J \otimes I^J$, where $U^J$ operates on the $\mathcal{H}_x^J$ component, with an identity $I^J$ on $\mathcal{K}_x^J$. Hence each $J$ component is operated upon independently, the spin eigenvalue $J$ being conserved.

In summary, therefore, the ignorant observer may engineer any thermal operation extracting work separately from each $J$ component (depicted in Fig. 2). We can think of their operations being conditioned on the spatial symmetry type, and although $J$ is observed to fluctuate randomly, a certain amount of work is extracted for each $J$ (see the latter part of the results section for a more detailed analysis of this fluctuation). For each $J$, there exists a free operation within the thermal operations framework[25] that performs deterministic work extraction saturating inequality (1). This is because the transformation is between (energy-degenerate) uniformly mixed states of differing dimension. Note that the work extraction process does not involve a measurement by the observer—only a coupling to the apparatus that depends on the value of $J$. Therefore there is no need to consider an additional entropic measurement cost, unlike the case of Maxwell's demon[1,37].

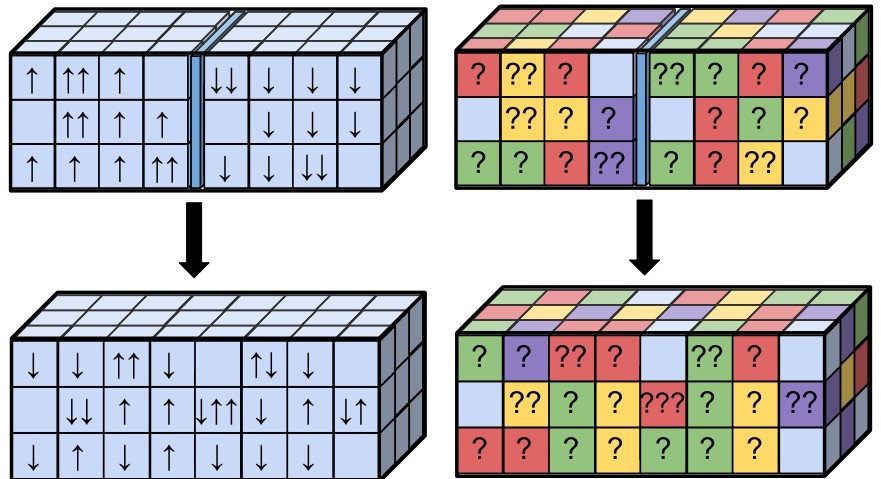

**Fig. 2 Schematic of the quantum mixing process.** Two diagrams representing the mixing of indistinguishable (bosonic) quantum gases from the perspective of the informed (left) and ignorant (right) observers. Initially, $n$ spin—↑ particles are found on the left and $m$ spin—↓ on the right. The particles are then allowed to mix while coupling to an external heat bath and work reservoir. The informed observer describes microstates via the number of particles in each cell, and their respective spins. The ignorant observer cannot tell the spins' states, but describes microstates (schematically depicted here by different colours) as superpositions of cell configurations, determined by the decomposition (6).

The question of optimal work extraction thus reduces to calculating the entropy of the initial state (8) and finding the maximum entropy final state. The fully thermalised final state seen by the ignorant observer is maximally mixed within each $J$ block:

$$\rho'_x = \bigoplus_J p_J \frac{I^J_x}{d_J} \otimes \mathrm{tr}_s |\phi_J\rangle\langle\phi_J|_{xs}, \tag{9}$$

where $I^J_x$ is the identity on $\mathcal{H}^J_x$ and $d_J$ is the corresponding dimension.

The overall entropy change is the average over all $J$, found to be (with details in Supplementary Note 2):

$$\Delta S_{\mathrm{igno}} = \sum_J p_J \Delta S^J_{\mathrm{igno}} \tag{10}$$

$$= \sum_J p_J \ln d^B_J - \ln\binom{n + d/2 - 1}{n} - \ln\binom{m + d/2 - 1}{m} \tag{11}$$

for bosons, and

$$\Delta S_{\mathrm{igno}} = \sum_J p_J \ln d^F_J - \ln\binom{d/2}{n} - \ln\binom{d/2}{m} \tag{12}$$

for fermions. Expressions for the dimensions $d^{B,F}_J$ are found in Supplementary Note 4:

$$d^B_J = \frac{(2J + 1)\left(\frac{N}{2} - J + d - 2\right)!\left(\frac{N}{2} + J + d - 1\right)!}{\left(\frac{N}{2} - J\right)!\left(\frac{N}{2} + J + 1\right)!(d - 1)!(d - 2)!}, \tag{13}$$

$$d^F_J = \frac{(2J + 1)d!(d + 1)!}{\left(\frac{N}{2} + J + 1\right)!\left(\frac{N}{2} - J\right)!\left(d - \frac{N}{2} + J + 1\right)!\left(d - \frac{N}{2} - J\right)!}. \tag{14}$$

The probabilities $p_J$ are found (see Supplementary Note 2) from the Clebsch-Gordan coefficients $C(j_1, m_1; j_2, m_2; J, M)$ describing the coupling of two spins with angular momentum quantum numbers $(j_1, m_1), (j_2, m_2)$ into overall quantum numbers $(J, M)$. Here, the two spins are the groups of particles on the left and right, respectively.

For identical gases, all particles have spins in the same direction, so the spin wavefunction is simply $|\uparrow\rangle^{\otimes N}$. This state lies fully in the subspace of maximal total spin eigenvalue, $J = M = N/2$—which is also fully symmetric with respect to permutations. Thus the spin part factorises out (i.e. there is no correlation between spin and spatial degrees of freedom). It is then clear that dimension counting reduces to the classical logic of counting ways to distribute particles between cells. Indeed, the dimension of the subspace $\mathcal{H}^{N/2}_x$ is $d^B_{N/2} = \binom{N + d - 1}{N}$ for bosons and $d^F_{N/2} = \binom{d}{N}$ for fermions. It follows that we recover the entropy as the classical case of indistinguishable particles (3).

For orthogonal spins, there are $n$ spin—↑ and $m$ spin—↓, leading to $M = (n - m)/2$ and a distribution over different values of $J$ according to

$$p_J = \frac{(2J + 1)n!m!}{\left(\frac{N}{2} + J + 1\right)!\left(\frac{N}{2} - J\right)!}. \tag{15}$$

The resulting entropies and significant limits are discussed after an example.

**Example**. Taking $n = m = 1$ demonstrates the mechanism behind the state space decomposition. For two particles, there are only two values of $J$, corresponding to the familiar singlet and triplet subspaces:

$$\begin{aligned}\mathcal{H}^0_s &= \mathrm{span}\{|\uparrow\downarrow\rangle - |\downarrow\uparrow\rangle\}, \\ \mathcal{H}^1_s &= \mathrm{span}\{|\uparrow\uparrow\rangle, |\downarrow\downarrow\rangle, |\uparrow\downarrow\rangle + |\downarrow\uparrow\rangle\}.\end{aligned} \tag{16}$$

Consider a spatial configuration where a spin—↑ particle is on the left in cell $i$, and a spin—↓ is on the right in cell $j$. For bosons, the properly symmetrised wavefunction is

$$|\psi_{i,j}\rangle := \frac{1}{\sqrt{2}}\left(|i_L j_R\rangle_x |\uparrow\downarrow\rangle_s + |j_R i_L\rangle_x |\downarrow\uparrow\rangle_s\right) \tag{17}$$

$$= \frac{1}{\sqrt{2}}\left[\frac{|i_L j_R\rangle - |j_R i_L\rangle}{\sqrt{2}} \cdot \frac{|\uparrow\downarrow\rangle - |\downarrow\uparrow\rangle}{\sqrt{2}} \quad (J = 0)\right.$$
$$\left. + \frac{|i_L j_R\rangle + |j_R i_L\rangle}{\sqrt{2}} \cdot \frac{|\uparrow\downarrow\rangle + |\downarrow\uparrow\rangle}{\sqrt{2}} \quad (J = 1)\right]. \tag{18}$$

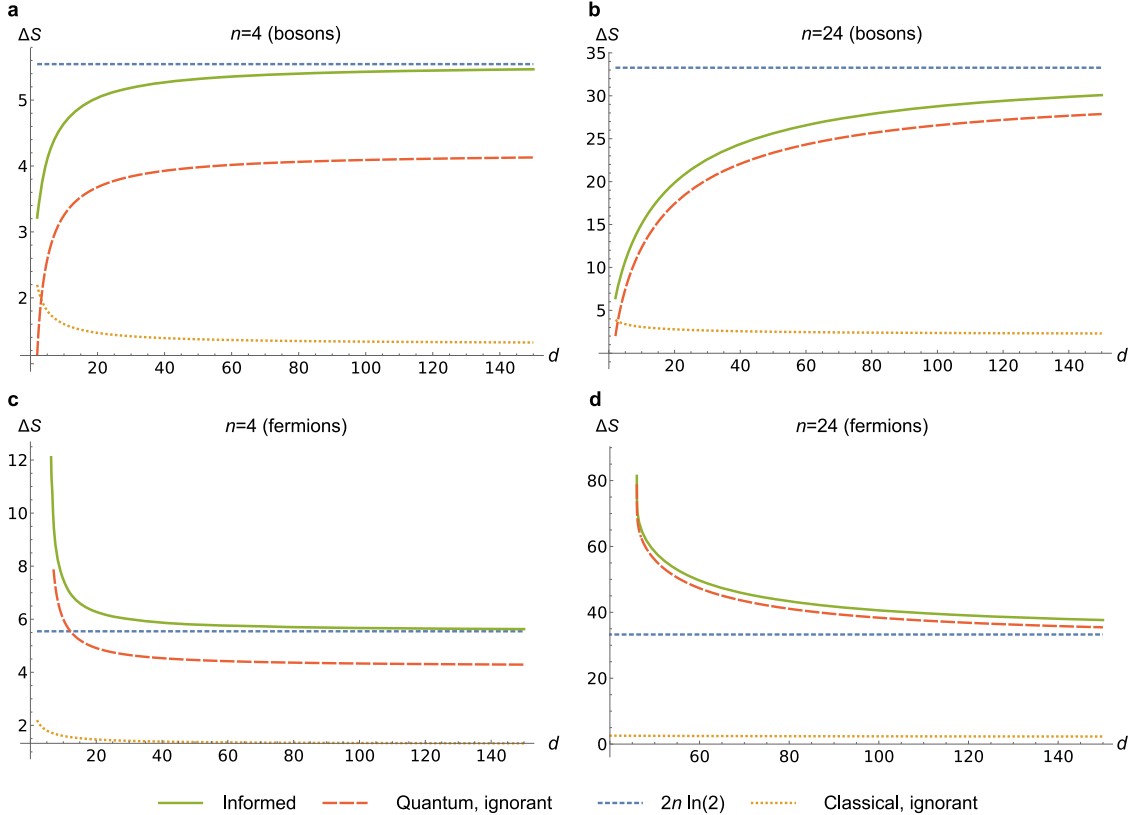

**Fig. 3 Entropy changes as a function of dimension.** Series of plots showing $\Delta S_{\text{info}}$, $\Delta S_{\text{igno}}$ against the total cell number $d$ of the system. **a**, **b** Bosonic systems of particle number $n = 4$ and $n = 24$ respectively. **c**, **d** The same for fermionic systems. Note that we have taken the initial number of particles on either side of the box to be equal, $n = m$ in all cases. For comparison, all four figures also display the classical changes in entropy for an informed/ignorant observer. The behaviour of the deficit between $\Delta S$ for an informed/ignorant observer of quantum particles agrees with the low density limit in Eq. (23) where we can see $\Delta S_{\text{info}}$ tending to the classical limit $2n \ln(2)$ with $\Delta S_{\text{igno}}$ trailing behind by a deficit of $n^2/2d^2 + H(\mathbf{p})$. Additionally, by comparing the different plots, we can see the low-dimensional fermionic advantage where the change in entropy is even greater than the classical $2n \ln 2$ value.

So $p_0 = p_1 = 1/2$, and the spatial component of this state is conditionally pure for both $J$. The initial thermal state is a uniform mixture of all such $|\psi_{i,j}\rangle$, with $(d/2)^2$ terms. Thus $S(\rho_x^0) = S(\rho_x^1) = 2(\ln d - \ln 2)$. For the final thermal state, we observe that

$$\mathcal{H}_x^0 = \text{span}\{|ij\rangle - |ji\rangle |i < j\}, \tag{19}$$

$$\mathcal{H}_x^1 = \text{span}\{|ij\rangle + |ji\rangle |i \leq j\}, \tag{20}$$

where $i, j$ now label cells either on the left or right. The corresponding dimensions are $d_0 = d(d-1)/2$, $d_1 = d(d+1)/2$. Within the $J = 0$ subspace, the entropy change is $\ln[d(d-1)/2] - 2\ln d + 2\ln 2 = \ln(1 - 1/d) + \ln 2$, and for $J = 1$, it is $\ln[d(d+1)] - 2\ln d + 2\ln 2 = \ln(1 + 1/d) + \ln 2$. Overall, therefore,

$$\Delta S_{\text{igno}} = \frac{1}{2}\ln\left(1 - \frac{1}{d}\right) + \frac{1}{2}\ln\left(1 + \frac{1}{d}\right) + \ln 2 \tag{21}$$

$$= \frac{1}{2}\ln\left(1 - \frac{1}{d^2}\right) + \ln 2. \tag{22}$$

For the informed observer, we have $\Delta S_{\text{info}} = 2\ln 2$. For identical gases, we find $\Delta S_{\text{iden}} = \ln(1 + 1/d) + \ln 2$, strictly greater than $\Delta S_{\text{igno}}$, but the two become equal in the limit $d \to \infty$.

Repeating the same calculation with fermions, the symmetric and antisymmetric states now pair up oppositely. Then $\Delta S_{\text{igno}}$ is the same as for bosons. However, we have

$\Delta S_{\text{iden}} = \ln(1 - 1/d) + \ln 2 < \Delta S_{\text{igno}}$. Unlike for bosons, two distinguishable fermions permit more extractable work by the ignorant observer than two identical fermions.

**Entropy changes and limits**. In Fig. 3 we plot both $\Delta S_{\text{info}}$ and $\Delta S_{\text{igno}}$ as a function of dimension for bosons and fermions. Below we analyse the special cases and limits which emerge from these expressions, summarised in Table 2.

With bosons, there are two special cases in which it is easily proven that distinguishable gases are less useful than indistinguishable ones for the ignorant observer. The first case is the example above, with $n = m = 1$. In addition, for $d = 2$, we have $d_J^B = 2J + 1$ —so the largest subspace is that with maximal $J = N/2$. The largest entropy change is then obtained when $p_{N/2} = 1$, which is satisfied precisely for indistinguishable gases.

For fermions, we see from Fig. 3 that the greatest work—for both observers—is obtained for small $d$. An intuitive explanation is that the Pauli exclusion principle causes the initial state to be constrained and thus have low entropy. For example, with minimal dimension $d = 2n = 2m$, we have $\Delta S_{\text{info}} = 2\ln\binom{2n}{n} \approx 4n\ln 2$ to leading order when $n$ is large. The ignorant observer can do almost as well: the state is entirely

**Table 2 Summary of results.**

| Limit | Quantum (no limit) | Classical (no limit) | Quantum ($d \gg n$) | Quantum ($d \gg n \gg 1$) | Classical ($d \gg n \gg 1$) |
|---|---|---|---|---|---|
| $\Delta S_{\text{info}}$ | $2\ln\binom{n+d-1}{n} - 2\ln\binom{n+d/2-1}{n}$ | $2\ln\binom{n+d-1}{n} - 2\ln\binom{n+d/2-1}{n}$ | ... | $\approx 2n\ln 2$ | $\approx 2n\ln 2$ |
| $\Delta S_{\text{igno}}$ | $\sum_J p_J \ln d_J^B - 2\ln\binom{n+d/2-1}{n}$ | $\ln\binom{2n+d-1}{2n} - 2\ln\binom{n+d/2-1}{n}$ | $\approx \Delta S_{\text{info}} - H(\mathbf{p})$ | $\approx 2n\ln 2$ | $\approx 0$ |

Entropy changes $\Delta S_{\text{info}}$, $\Delta S_{\text{igno}}$ for the informed and ignorant observers and their limits are expressed for bosons with $n = m$. For fermions, replace the dimension of the symmetric subspace $\binom{n+d-1}{n}$ with that of the antisymmetric one $\binom{d}{n}$ and $d_J^B$ by $d_J^F$ (both of which are defined in Eq. (13)).

contained in the $J = 0$ subspace, with $d_0^F = \frac{(2n)!(2n+1)!}{(n!)^2(n+1)!^2} = \frac{2n+1}{(n+1)^2}\binom{2n}{n}^2$, giving $\Delta S_{\text{igno}} \approx 4n\ln 2$ for large $n$. This is twice as much as for the classical ideal gas.

The most interesting conclusion is reached in the limit of large $d \gg n$, which we term the low density limit. For simplicity, we take $n = m$. We find

$$\Delta S_{\text{igno}} \approx \Delta S_{\text{info}} - H(\mathbf{p}), \qquad (23)$$

where $H(\mathbf{p}) = -\sum_J p_J \ln p_J$ is the Shannon entropy of the distribution $p_J$, and the lowest order correction is $-n^2/2d^2$. Thus, as $d \to \infty$, the ignorant observer can extract as much work as the informed one, minus an amount $H(\mathbf{p})$. This gap is evident from the graphs in Fig. 3.

Now consider the limit $d \gg n \gg 1$, with both low density and large particle number. Classically, this limit recovers ideal gas behaviour—the large dimension limit can be thought of as letting the box become a continuum. In Supplementary Note 6, we show that $H(\mathbf{p})$ (which depends only on $n$, not $d$), behaves as

$$H(\mathbf{p}) \approx \frac{1}{2}\ln n + 0.595..., \qquad (24)$$

with a correction going to zero as $n \to \infty$. Recall that the entropy change for the informed observer is approximately $2n\ln 2$ in this limit. Therefore the deficit $H(\mathbf{p})$, which is logarithmic, becomes negligible compared with $2n\ln 2$. Thus the ignorant observer can extract essentially as much work as the informed observer: $\Delta S_{\text{igno}} \approx \Delta S_{\text{info}} \approx 2n\ln 2$. This result is remarkable because it shows a macroscopic departure from the classical case in this limit.

How can we understand this low density limit? An important feature of the low density limit is that the final entropy becomes as large as it could possibly be: $\rho'_x$ becomes maximally mixed over its whole state space. This is true for any $N$, not just large numbers. We now give an explanation of this phenomenon, which proceeds by counting the number of mutually orthogonal states which can be accessed by the ignorant observer.

The important point about the low density limit is that particles almost never sit on top of each other—that is, almost all states are such that precisely $N$ cells are occupied, each with a single particle. More formally, the number of ways of putting $N$ bosonic particles into $d$ cells is $\binom{N+d-1}{N} \approx \binom{d}{N}$ when $d$ is large, where the approximation means the ratio of the two sides is close to unity. Let us refer to each of these $\binom{d}{N}$ choices of (singly) occupied cells as a cell configuration. For each cell configuration, there are $\binom{N}{n}$ spin configurations, i.e. ways of

distributing the $n$ spin—$\uparrow$ and $m$ spin—$\downarrow$ particles. In classical physics, the ignorant observer cannot distinguish any of the spin configurations corresponding to a single cell configuration. In quantum mechanics, remarkably, there are precisely $\binom{N}{n}$ states which can be fully distinguished by the ignorant observer, each being a superposition of different spin configurations.

Let us choose a single cell configuration—without loss of generality, let cells $1,...,N$ be occupied. The state of a spin configuration is denoted as a permutation of

$$|\uparrow\rangle_1 \cdots |\uparrow\rangle_n |\downarrow\rangle_{n+1} \cdots |\downarrow\rangle_N \in (\mathbb{C}^2)^{\otimes N}, \qquad (25)$$

where each cell is treated as a qubit with basis states $|\uparrow\rangle, |\downarrow\rangle$ according to which type of spin occupies it. (Note that the subsystems being labelled are here are the occupied cells, not particles.)

Again using Schur-Weyl duality, the state space of $N$ qubits can be decomposed as

$$(\mathbb{C}^2)^{\otimes N} = \bigoplus_J \mathcal{H}^J \otimes \mathcal{K}^J. \qquad (26)$$

Due to this decomposition, there is a natural basis $|J, M, p\rangle$, where SU(2) spin rotations $u_s^{\otimes N}$ act on the $M$ label (denoting the eigenvalue of the total $z$-direction spin), and permutations $\Pi$ of the $N$ cells act on the $p$ label.

How do we represent the effective state seen by the ignorant observer? In the representation used here, this corresponds to twirling over the spin states, i.e. performing a Haar measure average over all spin rotations $u_s^{\otimes N}$[38]. In the basis $|J, M, p\rangle$, however, this is a straightforward matter of tracing out the $\mathcal{H}^J$ subspaces, since only these are acted on by the twirling operation. Thus the ignorant observer has access to states labelled as $|J, p\rangle$.

How much information has been lost by tracing out $\mathcal{H}^J$? In fact, none—the label $M = (n-m)/2$ is fixed. Therefore the experimenter can perfectly distinguish all the basis states $|J, p\rangle$—and there are just as many of these as there are spin configurations, namely $\binom{N}{n}$.

For example, take $n = m = 1$: the two spin configurations are $|\uparrow\downarrow\rangle$, $|\downarrow\uparrow\rangle$, and for some pair of occupied cells, the two distinguishable states are

$$|J = 1, M = 0, p = 0\rangle = \frac{1}{\sqrt{2}}(|\uparrow\downarrow\rangle + |\downarrow\uparrow\rangle), \qquad (27)$$

$$|J = 0, M = 0, p = 0\rangle = \frac{1}{\sqrt{2}}(|\uparrow\downarrow\rangle - |\downarrow\uparrow\rangle). \qquad (28)$$

Since these are respectively in the triplet and singlet subspaces, they remain orthogonal even after twirling. They can be distinguished by mixing the cells at a balanced beam splitter: it is easy to show that the symmetric state ends up with a

superposition of both particles in cell 1 and both in cell 2, while the antisymmetric state ends up with one particle on each side. Therefore, after this beam splitter, the two states can be distinguished by counting the total particle number in each cell.

A slightly more complex example is with $n = 2, m = 1$. Then the distinguishable basis states for three occupied cells are

$$\left| J = \frac{3}{2},\ M = \frac{1}{2},\ p = 0 \right\rangle = \frac{1}{\sqrt{3}}(|{\uparrow\uparrow\downarrow}\rangle + |{\uparrow\downarrow\uparrow}\rangle + |{\downarrow\uparrow\uparrow}\rangle),$$

(29)

$$\left| J = \frac{1}{2},\ M = \frac{1}{2},\ p = 0 \right\rangle = \frac{1}{\sqrt{2}}(|{\downarrow\uparrow\uparrow}\rangle + |{\uparrow\downarrow\uparrow}\rangle),$$

(30)

$$\left| J = \frac{1}{2},\ M = \frac{1}{2},\ p = 1 \right\rangle = \sqrt{\frac{2}{3}}|{\uparrow\uparrow\downarrow}\rangle - \frac{1}{\sqrt{6}}(|{\uparrow\downarrow\uparrow}\rangle + |{\downarrow\uparrow\uparrow}\rangle).$$

(31)

Observe that the argument in this section does not depend in anyway on the exchange statistics of the particles, explaining why we see the same limit for bosons and fermions.

**Quantumness of the protocol**. The above discussion of the low density limit clarifies the fundamental reason why the quantum ignorant observer performs better than the classical one. The distinguishable states comprising the final thermalised state are superpositions of different spin configurations. We might describe a classical observer within the quantum setting as one who is limited to operations diagonal in the basis of cell configurations—that is, they are only able to count the number of particles occupying each cell. For such an observer, these superposition states are indistinguishable.

A crucial question is then: how difficult is it to engineer the quantum protocol for the ignorant observer? We can imagine that the heat bath and work reservoir might naturally couple to the system in the cell occupation basis (if this is the basis that emerges in the classical case). The required coupling is in the Schur basis $|J, q\rangle_x$, which are generally highly entangled between cells. A sense of their complexity is given by the unitary that rotates the Schur basis to the computational basis, known as the Schur transform. Efficient algorithms to implement this transform have been found[39], with a quantum circuit whose size is polynomial in $N$, $d$, $\ln(1/\epsilon)$, allowing for error $\epsilon$. This circuit is related to the quantum Fourier transform, an important subroutine in many quantum algorithms. Thus, while the Schur transform can be implemented efficiently, it appears that engineering the required work extraction protocol—in the absence of fortuitous symmetries in the physical systems being used—may be as complex as universal quantum computation.

**Work fluctuations**. The work extraction protocol we have presented is not deterministic: for each value of $J$, a different amount of work is extracted with probability $p_J$. This is typically expected of thermodynamics of small systems; however, in classical macroscopic thermodynamics, such fluctuations are negligible. We can ask whether the same is true of the work extracted by the ignorant observer in the quantum case, especially in the low density and large particle number limits.

One informative way of quantifying the fluctuations is via the variance of entropy change. Let us denote the entropy change for each $J$ by $\Delta S_{\text{igno}}(J)$. The mean is $\Delta S_{\text{igno}} = \sum_J p_J \Delta S_{\text{igno}}(J)$, and the variance is $V(\Delta S_{\text{igno}}) = \sum_J p_J \Delta S_{\text{igno}}(J)^2 - \Delta S_{\text{igno}}^2$. This can be computed straightforwardly from our expressions for $p_J$, $d_J$, and approximated in various limits.

Consider first a high density BEC-limit case with $d = 2$ and $N = 2n \gg 1$ bosons. We have $d_J^B = 2J + 1$, and using the techniques of Supplementary Note 6, $p_J \approx \frac{2J}{n} e^{-J^2/n}$. Then $\Delta S_{\text{igno}} = \sum_J p_J \ln(2J + 1) \approx \frac{1}{2}\ln n + \ln 2 - \frac{\gamma}{2} \approx \frac{1}{2}\ln n + 0.405$. Similarly, we compute $V(\Delta S_{\text{igno}}) = \sum_J p_J [\ln(2J + 1)]^2 \approx \frac{\pi^2}{24} \approx 0.411$. Therefore the mean work dominates its fluctuations (logarithmic versus a constant).

Next, consider the closest analogue for fermions: the case of minimal dimension $d = 2n = 2m$. Recall that $\Delta S_{\text{igno}} \approx \Delta S_{\text{info}} \approx 4n \ln 2$ for large $n$. Since $p_0 = 1$, work extraction is in fact completely deterministic in this case.

Finally, take the low density limit. As found before, for both bosons and fermions, $\Delta S_{\text{igno}} \approx 2n \ln 2$ —linear in $n$—and yet we still find a constant $V(\Delta S_{\text{igno}}) \approx \frac{\pi^2}{24}$.

In these macroscopic limits, therefore, work extraction is either fully deterministic or effectively deterministic in that the fluctuations are negligible compared with the mean.

**Non-orthogonal spins**. The results generalise to the case of partially distinguishable spins—that is, initially with $n$ in spin state $|{\uparrow}\rangle$ on the left and $m$ in state $|{\nearrow}\rangle$ on the right, where

$$|{\nearrow}\rangle = \cos(\theta/2)|{\uparrow}\rangle + \sin(\theta/2)|{\downarrow}\rangle.$$

(32)

For this, we must be more explicit about the operations permitted by the informed observer. The most general global unitary that does not affect the number of each type of spin is of the form $U = \bigoplus_M U_{xsBW}^{(M)}$, where the block structure refers to subspaces with fixed $M$ as defined by the Schur basis (recalling that the total number of particles is fixed). We find (see Supplementary Note 3 for details) that $\Delta S_{\text{info}}$ is an average of entropy changes for each value of $M$. For $\Delta S_{\text{igno}}$, all that changes is the probability $p_J$, now being obtained by an average over Clebsch-Gordan coefficients. Importantly, for both observers, the result is a function of $\theta$ only via the probability distribution $q_M$ for the spin value $M$. In Fig. 4, one observes the smooth transition from identical to orthogonal spin states as $\theta$ varies from 0 to $\pi$.

## Discussion

In contrast to the classical Gibbs paradox setting, we have shown that quantum mechanics permits the extraction of work from apparently indistinguishable gases, without access to the degree of freedom that distinguishes them. It is notable that the lack of information about this 'spin' does not in principle impede an experimenter at all in a suitable macroscopic limit with large particle number and low density—the thermodynamical value of the two gases is as great as if they had been fully distinguishable.

The underlying mechanism is a generalisation of the famous Hong–Ou–Mandel (HOM) effect in quantum optics[34,40,41]. In this effect, polarisation may play the role of the spin. Then a non-polarising beam splitter plus photon detectors are able to detect whether a pair of incoming photons are similarly polarised. The whole apparatus is polarisation-independent and thus accessible to the ignorant observer. Given this context, it is therefore not necessarily surprising that quantum Gibbs mixing can give different results to the classical case. However, the result of the low density limit is not readily apparent. This limit is reminiscent of the result in quantum reference frame theory[38] that the lack of a shared reference frame presents no obstacle to communication given sufficiently many transmitted copies[42].

Two recent papers[18,43] have studied Gibbs-type mixing in the context of optomechanics. A massive oscillator playing the role of a work reservoir interacts with the photons via their pressure. This oscillator simultaneously acts as a beam splitter between

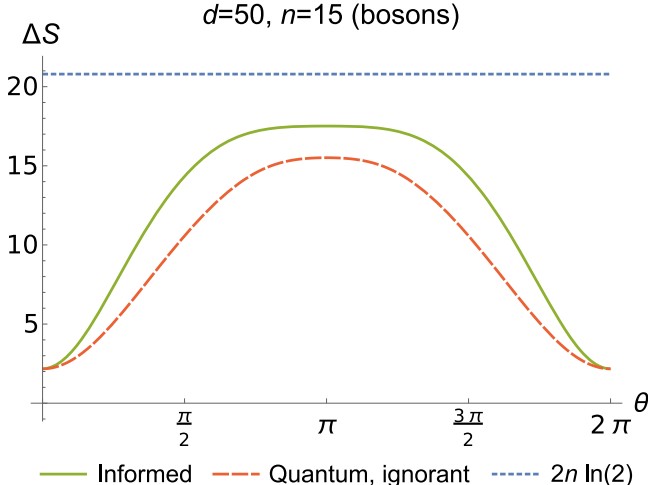

**Fig. 4 Results for partially distinguishable spins.** Plots of $\Delta S_{info}$, $\Delta S_{igno}$ as a function of orthogonality of the spin states as determined by $\theta$ in (32). The figure is for a bosonic system with initial numbers of particles on either side of the box $n = m = 15$, and $d = 50$ cells. For comparison, the figure also displays the classical change in entropy, $2n \ln (2)$. Here, the greatest change in entropy occurs when the spin states are orthogonal at $\theta = \pi$.

the two sides of the cavity. In ref. [18], the beam splitter is non-polarising and thus (together with the interaction with the oscillator) accessible to the ignorant observer. The main behaviour there is driven by the HOM effect, which enhances the energy transfer to the oscillator, albeit in the form of fluctuations. In ref. [43], which studies Gibbs mixing as a function of the relative polarisation rotations between left and right, bosonic statistics are therefore described as acting oppositely to Gibbs mixing effects—which is different from our conclusions. However, there is no contradiction: we have shown that an advantage is gained by optimising over all allowed dynamics. Moreover, the scheme in ref. [43] uses a polarisation-dependent beam splitter, which is only accessible to the informed observer. Therefore the effect described here cannot be seen in such a set-up. It is an interesting question whether such proposals can be modified to see an advantage of the type described here, even if not optimal.

It is important to determine how the thermodynamic enhancements predicted in this paper may have implications for physical systems. Such an investigation should make use of more practical proposals (such as refs. [16,18,43]) to better understand possible realisations of mixing. For example, systems of ultra-cold atoms in optical lattices[44] may provide a suitable platform to experimentally realise the thermodynamic effects predicted in this work. The question of the maximal enhancement in the macroscopic limit is particularly compelling given the rapid progress in the manipulation of large quantum systems[45].

## Methods
The Supplementary Information contains detailed proofs. Supplementary Note 1 describes the treatment of classical particles, starting from a description akin to first quantisation, and then coarse-graining the state space along with appropriate restrictions on the allowed dynamics. Supplementary Note 2 fills out the derivation for the quantum ignorant observer sketched in the main text. Supplementary Note 3 provides details for the general case of non-orthogonal spins. Supplementary Note 4 computes the dimensions of the spaces $\mathcal{H}_x^\lambda$ from representation theory formulas. Supplementary Notes 5, 6 show how to take the low density and large particle number limits, respectively.

## Code availability
Source code for generating the plots is available from the authors upon request.

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

## Acknowledgements

We acknowledge financial support from the European Research Council (ERC) under the Starting Grant GQCOP (Grant No. 637352), the EPSRC (Grant No. EP/N50970X/1) and B.M.'s doctoral prize EPSRC grant (EP/T517902/1). B.Y. is also supported by grant number (FQXi FFF Grant number FQXi-RFP-1812) from the Foundational Questions Institute and Fetzer Franklin Fund, a donor advised fund of Silicon Valley Community Foundation. We are grateful to Zoë Holmes, Gabriel Landi, Vlatko Vedral, Chiara Marletto, Bernard Kay, Matthew Pusey, Alessia Castellini and Felix Binder for valuable discussions.

## Author contributions

B.Y. and B.M. contributed equally to the research and writing the paper, supervised by G.A.

## Competing interests

The authors declare no competing interests.
