## [Peer Review File · Nature Communications]

Reviewers' Comments:

Reviewer #1:

Remarks to the Author:

This paper considers a toy model of Gibbs mixing: a box consisting of two halves (each half "gridded" into $d/2$ cells) that are initially each filled with a gas, and with the end state being one where each gas is expanded across the entire box.

(Coupling to a heat bath/work storage system is also discussed but this is something that didn't become clear to me from the main text.)

The paper discusses the combinatorial (Boltzmann/Gibbs) entropies of the initial and final states of the gases for three gas+observer situations:

- 1 - the two gases are different, the observer sees them as different
- 2 - the two gases are different, the observer sees them as identical
- 3 - the two gases are identical, the observer sees them as identical

The difference is encoded in spin states being up / down. The sole quantum aspect in the discussion is the particle statistics, which is either bosonic or fermionic, and that leads to either symmetrized or anti-symmetrized wave functions in the spin-position Hilbert space, i.e. instead of the full Hilbert space, bosons/fermions live in subspaces that have the correct symmetrization property under particle exchange.

When the observer can't see the spin-state difference of the two gases, case 2, they will associate a different entropy to the situation than case 1. The authors establish the set of different combinatorial configurations available for bosons/fermions and evaluate the entropies.

Classically, it is well known that work can be extracted from mixing the gases in case 1 while no work can be extracted in case 3.

The title hints that the paper's surprising result is that case 2 actually does give a work output, despite the fact that from the observer's point of view case 2 looks like case 3.

The paper is conceptually interesting and certainly publishable. It is less clear to me that Nature Comms is the right platform for the paper. The toy nature of the modelling as a combinatorial problem, see Table I, make the conclusions somewhat disconnected from a realistic situation.

Further, I have a few questions:

1. "Work" is defined in (1) as proportional to entropy differences but this presupposes that a process can be found that is optimal - otherwise the work is only *bounded* by the entropy difference. It may be clearer to abandon the mentioning of "work" in the paper and rather discuss "entropy changes", as this is really the paper focus. Connection to the Gibbs/Boltzmann entropy discussion may then be appropriate.

Alternatively a much more in-depth discussion of the impact of the presence of the heat bath/work storage system on the work extracted from the gases will be needed, as well as a discussion of how to optimally achieve the maximal work output.

2. Given the combinatorial approach to entropies, temperature does not seem to feature. It only seems to be included as a prefactor in (1). The authors should discuss this and where the toy example lies with respect to the microcanonical/canonical ensemble picture. Furthermore, as well known for bosons, a linear temperature dependence of energies is common in the canonical picture in the high temperature limit — but at low temperatures energies often scale with a coth in inverse temperature. Despite the Bosonic/Fermionic nature of the gases, this seems to not be taken into account here, which appears inconsistent.

3. Can one use the toy model to discuss gases with spin states that are NOT orthogonal? I.e. the quantum case beyond particle statistics?

4. At small d , the bosonic/fermionic symmetrisation requirement implies entanglement between spatial modes, a topic widely discussed about a decade ago. Can deviations between the finite- d - ΔS and the high- d -limit- ΔS be linked to an entanglement entropy/measure?

5. I don't understand why the authors say their results are surprising in the low density limit, d =large. Here they conclude that ΔS is the same ($2n \ln 2$) for cases 1 and 2 — but that would be what one would macroscopically expect, no? What the observer knows should not impact on the entropy change in the classical/macro world.

I may have misunderstood this point. In that case it'd be good for the authors to better list/label the 3 situations discussed, eg. as cases 1, 2, and 3 as above. The current labelling of info and info, the distinction between what is the initial and final state in the respective cases, and the distinction between the classical and quantum cases is not fully clear to me.

6. It's surprising that no comparison of the setup/results is made with those of two recent papers on the same topic. One discussing the energy transfer due to the Hong-Ou-Mandel effect for polarisation [18] — mentioned without the reference in the paper's conclusion. Another on "quantum Gibbs mixing" [42] — i.e. same key words in the title as this paper -- is mentioned only in a footnote. The authors should address eg in intro/discussion how the setup/results of the present paper compare with those, or if they are not comparable why not?

7. In the conclusion mentioning how the results here are important for heat engines seems misleading to me. To be able to close a heat engine cycle would mean reversing the mixing?!; that's not something usually considered. I.e. if the authors want to imply that their results can make a difference to heat engine cycles, then they should detail that a bit more.

Reviewer #2:

Remarks to the Author:

The present manuscript re-examines the Gibbs paradox for mixing indistinguishable quantum particles. It is found that in contrast to the classical case the so-called ignorant observer can extract as much work (in the macroscopic limit) as the informed observer. This genuine quantum effects originates in carefully accounting for distinguishable quantum states, which is not identical to counting classically distinguishable microstates.

Quantum thermodynamics is the study of the thermodynamic properties of quantum systems. Although the phenomenological framework remains unchanged, surprising and subtle quantum effects may be sought in carefully computing thermodynamic quantities from the underlying quantum states. The present manuscript does just that in the context of the Gibbs paradox. The result is interesting, and the analysis is transparent. The narrative is pleasant to read, and the discussion is convincing. Therefore, I would like to see the manuscript eventually be published in Nature Communications.

However, I have several conceptual issues that the authors may find interesting to address in their revisions:

(i) Carefully thinking about Gibbs's paradox it seems apparent that the problem of mixing (in-)distinguishable gases is a generalization of Maxwell's demon. Classically, more work can be extracted by the "informed observer", that is by the observer that has taken a measurement with respect to the nature of the gases. The natural question arises how to rephrase the present arguments in terms of proper "quantum observations". In order to distinguish the quantum states (to get the counting right), one has to actually measure the system. The question then is whether one needs to more carefully account for the "cost" of the measurement and the potential quantum back action on the gases' states.

(ii) The authors assign a spin degree of freedom to the gas particles, which constitutes a label of distinguishability. From a more thermodynamic point of view this spin is nothing else but an "information bearing degree of freedom". Thus, a concise discussion in the terminology of (quantum) thermodynamics of information might help to make the conceptual part of the analysis a lot more accessible.

(iii) The authors introduce the notion of a "work battery". Since this is a notion exclusive to the resource theoretic framework to quantum thermodynamics, I wonder why this terminology is used in the present context. The analysis is phrased entirely in a standard language of statistical mechanics, and hence the term "work reservoir" may be more apt.

(iv) In Eq. (4) the authors find the extractable work as difference of the NONEQUILIBRIUM free energies. It is easy to see that the system is initially prepared in a state of local equilibrium, which relaxes towards a state of global equilibrium. Hence, a more careful formulation of the thermodynamics is crucial. The considered work is actually the "excess work", which accounts for the full non equilibrium character of the situation. Also, it may be worthwhile to add a citation for the nonequilibrium free energy, which to the best of my knowledge was introduced in

Bergmann & Lebowitz, PR 99, 578 (1955)

Reviewer #3:

Remarks to the Author:

In this manuscript the authors analyze the Gibbs paradox -- involving the entropy change upon mixing apparently indistinguishable particles -- within a quantum framework. They cast the problem by imagining a gas of spin-up particles initially confined to the left half of a box, and a gas of spin-down particles initially confined to the right half of a box. Assuming (for simplicity) equal densities on both sides, what is the change in the entropy of the system when a partition dividing the two sides is removed? In the classical case, an "informed" observer able to distinguish spin-up from spin-down particles, and an "ignorant" observer unable to do so, would answer this question differently. Their answers reflect the difference in the maximum amount of work extractable by either observer, in a process that takes the system from the initial (unmixed) to the final (mixed) state.

Analyzing the quantum case in detail, the authors take care to account for the symmetry properties imposed by quantum mechanics, related to whether the particles are bosons or fermions -- both cases are analyzed. The symmetry or anti-symmetry of the many-particle wavefunction effectively creates non-classical entanglements, which have important consequences for the entropy change calculated by the ignorant observer. (In the case of the informed observer, the entropy change in the quantum case is the same as in the classical case.) The authors analyze this problem using Schur-Weyl duality, and show that the entropy change for the ignorant observer, in the quantum case, is substantially larger than in the classical case. Indeed, in the "macroscopic" limit of low density and large particle number, they argue that the entropy change as determined by the ignorant observer in the quantum case, approaches the entropy change determined by the informed observer in the quantum case! To put it differently: in the macroscopic limit an ignorant observer (unable to distinguish spin-up from spin-down particles) can extract as much work from the mixing process as an informed observer.

These results are both counter-intuitive and interesting, and they add to our understanding of the Gibbs paradox. The analysis in the paper is not straightforward to follow, as it involves somewhat specialized theoretical tools such as Schur-Weyl duality and the Schur basis. Nevertheless, I believe the analysis is correct, and that a careful reader can understand the meaning of the central result of the paper, even if that reader does not follow every step of the analysis.

For these reasons I believe this manuscript deserves publication in Nature Communications.

Reply to referees for NCOMMS-20-34210:

Mixing indistinguishable systems leads to a quantum Gibbs paradox

To summarise the main points addressed in our reply:

In response to the first referee, we have clarified the achievability of the optimal extracted work, improved presentation of the different cases considered in the manuscript, and added a comparison with recent papers on a similar topic. We have also added substantial material to solve the more general case of non-orthogonal spins, which permits a continuous interpolation between identical and distinguishable gases. We hope that our explanations make clear the case for publication in this journal – especially why the found effect is so counter-intuitive from a classical standpoint, and why this is an important conceptual advance that would benefit from high visibility.

In response to the second referee, we have clarified why there are no hidden measurement costs, and employed more standard terminology to put our work in context within the broader formulations of thermodynamics.

Reply to Referee #1

1. *The toy nature of the modelling as a combinatorial problem, see Table I, make the conclusions somewhat disconnected from a realistic situation.*

We thank the Referee for this observation. While we agree that the toy model is not something feasibly implementable in the near term, we believe that our approach is still particularly valuable. Firstly, our analysis can be regarded as a quantum version of the seminal combinatorial state-counting arguments studied by Boltzmann [1] which are known to recover the entropy changes for the ideal gas [2–4]. Such a toy model is thus suitable to reveal the important points of the argument in the classical case. Our approach uncovers the important conceptual leap that is missed by these textbook classical approaches. Moreover, it distills the problem down to its core essential elements so that we can highlight the cause of the novel effect without unnecessary complications. Finally, it enables us to derive fundamental limits on the effect in terms of optimal entropy changes. These limits will remain of relevance as general benchmarks even in more practical settings.

We also agree with the Referee that it is important to find more realistic proposals for observing the effect in experiments – this is highly interesting but will be a substantial piece of follow-up work that lies beyond the scope of the present paper. We have now commented on such an outlook in the conclusions.

2. *“Work” is defined in (1) as proportional to entropy differences but this presupposes that a process can be found that is optimal - otherwise the work is only *bounded* by the entropy difference. It may be clearer to abandon the mentioning of “work” in the paper and rather discuss “entropy changes”, as this is really the paper focus. Connection to the Gibbs/Boltzmann entropy discussion may then be appropriate. Alternatively a much more in-depth discussion of the impact of the presence of the heat bath/work storage system on the work extracted from the gases will be needed, as well as a discussion of how to optimally achieve the maximal work output.*

We thank the Referee for this valuable comment and we have now endeavoured to make the connection between entropy changes and useful mechanical work clearer in the text. We do refer to work in order to emphasise that the effect has measurable thermodynamical implications – some readers might otherwise have dismissed “abstract entropies” as not directly physically relevant.

As the Referee correctly points out, the equality rather than inequality in equation (1) assumes that a process taking $\rho \rightarrow \rho'$ while optimally coupling to a heat bath/work reservoir exists; any non-optimal process results in some useless heat lost to the environment. Deriving the explicit energy preserving unitary which couples the system/bath/reservoir tends to be a nontrivial task in itself and often physically depends on external limitations provided by the experimental setup. However, what is known from the resource theoretic treatment of thermodynamics is that such a unitary extracting the quoted amount of work definitely exists [5, 6]. This is in the same sense that when one talks about work extraction classically, one often states that there exists an adiabatic transformation that indeed saturates the bound imposed by the 1st law.

In order to make the above clearer we have replaced the equality in equation (1) with an inequality representing the bound imposed by the 1st law and devoted the text preceding and following this equation to how one might optimally saturate this bound both classically and with quantum systems.

The extractable work in a process that takes ρ to ρ' is then

$$W \leq F(\rho) - F(\rho') = k_B T [S(\rho') - S(\rho)]. \quad (1)$$

In a classical reversible process, the extractable work is equal to the change in free energies. This is generally an over-simplification for small systems, in which work can be defined in various ways [7] – e.g. required to be deterministic in the resource theory context [6] or as a fluctuating random variable [8, 9], requiring consideration of other varieties of free energy. However, equation (1) will turn out to be sufficient for our purposes in the sense of mean extractable work. We find the inequality to be saturable using thermal operations and characterise fluctuations around the mean in the latter part of our results section.

When outlining the protocol performed by the ignorant observer, we have also added the following:

For each J , there exists an operation within the thermal operations framework [6] that performs deterministic work extraction saturating inequality (1). This is because the transformation is between (energy-degenerate) uniformly mixed states of differing dimension.

It should also be noted that in response to Referee 2 we have also made changes to the paper that should make this discussion clearer.

3. Given the combinatorial approach to entropies, temperature does not seem to feature. It only seems to be included as a prefactor in (1). The authors should discuss this and where the toy example lies with respect to the microcanonical/canonical ensemble picture. Furthermore, as well known for bosons, a linear temperature dependence of energies is common in the canonical picture in the high temperature limit but at low temperatures energies often scale with a coth in inverse temperature. Despite the Bosonic/Fermionic nature of the gases, this seems to not be taken into account here, which appears inconsistent.

The Referee is correct that temperature effectively only appears as a prefactor in (1). The reason for this is the zero system Hamiltonian. Therefore, all appearances of temperature in our manuscript correspond to the temperature of the bath that the system interacts with. Of course this affects what you can do with the system, such as the quantity of extractable work – this is why the temperature factor appears in the free energy.

The temperature-dependent energies mentioned by the Referee are presumably the mean energy etc. resulting from the grand canonical picture – i.e., when particles are exchanged with a reservoir and only the mean particle number is fixed. Since the box is closed, we do not allow for such fluctuations. Moreover, these energies cannot be temperature-dependent again because we have $H = 0$. Due to the energy degeneracy, one could include our model in either the canonical or micro-canonical picture.

In order to make this clearer in the manuscript we have added the following text.

Since the particle number is strictly fixed, we are working in the canonical ensemble (rather than the grand canonical ensemble).

Hopefully, the above discussion of the canonical picture that applies to our bosonic and fermionic systems should also make it clear why we see no such quoted energetic dependence on temperature. Such factors will most likely be relevant in future work with more specific models.

4. Can one use the toy model to discuss gases with spin states that are NOT orthogonal? I.e. the quantum case beyond particle statistics?

We thank the Referee for the interesting question – this is something we had previously considered but not managed to solve. Thankfully, in the new manuscript we do have an answer. The results are summarised at the end of the main body of text, with technical details in Supplementary Note 3. For both observers, the entropies are essentially obtained by averaging over different values of M , the spin eigenvalue along the z -axis, which is due to the fact that there is now an uncertain number of spin-up and spin-down particles. Newly incorporated Fig. 4 plots these entropies against the relative spin orientation angle θ , for a certain choice of particle numbers and dimensions. As one might expect, there is a smooth interpolation between the minimum for identical spins ($\theta = 0$) and the maximum for orthogonal spins ($\theta = \pi$).

We would not necessarily say that generalising to non-orthogonal spins gives a case *beyond* particle statistics for our mixing phenomenon, as the most extreme effect is indeed found for orthogonal spins. However, this addition shows that our toy model is versatile enough to allow for a meaningful analysis of the case of generally non-orthogonal spin states.

5. At small d , the bosonic/fermionic symmetrisation requirement implies entanglement between spatial modes, a topic widely discussed about a decade ago. Can deviations between the finite- d -Delta S and the high- d -limit-Delta S be linked to an entanglement entropy/measure?

This is an interesting point brought up by the Referee, and one which we did consider when preparing our manuscript: whether the entanglement between the particles due to symmetrisation [10, 11] is the property responsible for some of the effects we see in our results.

It should be initially made clear that the bosonic/fermionic symmetrisation doesn't technically create entanglement between spatial modes but rather can create entanglement between the particles in the first-quantised picture (this particle entanglement can then be "activated" into spatial mode entanglement [10, 11]). In addition, this effect is not something which is limited to a system with a small number of spatial degrees of freedom but can and does happen in both the small and large d limits.

What is definitely true is that one main property responsible for our main result, that an ignorant quantum observer can still extract work, is due to the correlations between the spatial and spin

parts of the wavefunction. Hence, when an ignorant observer has no access to the spin part of the wavefunction they can still infer details of the lost information in the accessible spatial part.

However, it is also true that even with the same spins on each side of the box, the system would *still contain entanglement* between the particles via their spatial degrees of freedom. Therefore it doesn't appear to be the resource directly responsible for the quantum advantage in our results.

The type of entanglement that is really important for our effect is that outlined in the sections titled "Explaining the low density limit" and "Quantumness of the protocol": the spatial mode entanglement of the Schur basis states. These are states of the position degree of freedom which encode more information than would be accessible to the classical ignorant observer. In some sense it is obvious that the mode entanglement in these states is crucial: e.g. the states in eqn (22) cannot be discriminated by purely local measurements. However, it is difficult at present to be more quantitative than this since the Schur basis states do not have straightforward analytic expressions.

*6. I dont understand why the authors say their results are surprising in the low density limit, $d=\text{large}$. Here they conclude that ΔS is the same ($2n \ln 2$) for cases 1 and 2 but that would be what one would macroscopically expect, no? What the observer knows should not impact on the entropy change in the classical/macro world. I may have misunderstood this point. In that case itd be good for the authors to better list/label the 3 situations discussed, eg. as cases 1, 2, and 3 as above. The current labelling of *igno* and *info*, the distinction between what is the initial and final state in the respective cases, and the distinction between the classical and quantum cases is not fully clear to me.*

It is important to re-emphasise that, in the same vein as Jaynes, we have equated an observer's ability to learn information about the system with their ability to control it. This provides the resolution to the classical paradox.

The classical expectation, that surprisingly doesn't hold in the quantum case, is that the *ignorant* observer cannot extract more work from different gases than from identical gases. I.e., one would classically expect that an observer who lacks information about an internal degree of freedom cannot tell the difference between identical and orthogonal spins.

As the classical Gibbs paradox shows, it is *not* the case that "what the observer knows should not impact on the entropy change". The effect of knowledge on the entropy (or extractable work) is given by the difference between the two observers' results.

The surprising result here is that a quantum ignorant observer *can* extract work from gases with different spins, even without access to the spins. In the large d limit, this effect does not go away, but in fact becomes maximal. This is entirely opposite to classical expectations. It also demonstrates that we must not equate "classical" with "macroscopic". Our macroscopic limit is most definitely a non-classical effect. This challenges the expectation that one might recover classical thermodynamics from a simple limit of this type. In addition to the mixing effect we have studied, we believe this to be an important conceptual point in favour of publication in a high-visibility journal.

We appreciate that it is not straightforward to keep track of the different cases and so have included a summary table in the revised manuscript which the reader can easily refer to. We prefer not to replace the figure labels with, e.g. 1,2,3, since the meaning of these indices is not immediately apparent. We hope these additions and amendments make our manuscript easier to follow by interested readers.

7. It's surprising that no comparison of the setup/results is made with those of two recent papers on the same topic. One discussing the energy transfer due to the Hong-Ou-Mandel effect for polarisation

[18] mentioned without the reference in the papers conclusion. Another on “quantum Gibbs mixing” [42] i.e. same key words in the title as this paper – is mentioned only in a footnote. The authors should address eg in intro/discussion how the setup/results of the present paper compare with those, or if they are not comparable why not?

We thank the Referee for this meaningful comment. We have now devoted more space to the important comparison with these other papers. In the Discussion section, we have added:

Two recent papers [12, 13] have studied Gibbs-type mixing in the context of optomechanics. There, a massive oscillator playing the role of a work reservoir interacts with the photons via their pressure. There is also a beam splitter between the two sides of the cavity. In Ref. [12], the beam splitter is non-polarising and thus (together with the interaction with the oscillator) accessible to the ignorant observer. The main behaviour there is driven by the HOM effect, which impedes energy transfer to the oscillator. However, this does not contradict our findings: we have shown that an advantage is gained by optimising over all allowed dynamics. It is therefore an interesting question whether such proposals can be modified to see an advantage of the type described here, even if not optimal. Ref. [13] studies Gibbs mixing as a function of the relative polarisation rotations between left and right. However, this uses a polarising beam splitter, which is only accessible to the informed observer. Therefore the effect described here cannot be seen in such a set-up.

8. In the conclusion mentioning how the results here are important for heat engines seems misleading to me. To be able to close a heat engine cycle would mean reversing the mixing?!; thats not something usually considered. I.e. if the authors want to imply that their results can make a difference to heat engine cycles, then they should detail that a bit more.

We appreciate the Referee’s comment that at present it is not perfectly clear how to build a heat engine cycle which uses the fundamental effect discovered in this paper. We did not want to imply that this was a goal of our paper, more that it is a relevant naturally arising question – it would certainly involve substantial follow-up work. The modified part of the Discussion section now reads

It is important to determine how the thermodynamic enhancements predicted in this paper may have implications for physical systems. Such an investigation should make use of more practical proposals (such as Refs. [12–14]) to better understand possible realisations of mixing. For example, systems of ultra-cold atoms in optical lattices [15] may provide a suitable platform to experimentally realise the thermodynamic effects predicted in this work.

Incidentally, as the Referee states, when discussing reversible cycles in thermodynamics we must always close the cycle – but there should be no particular objection to reversing the mixing. As per the standard discussion in the classical setting, the informed observer can use a pair of semi-permeable membranes attached to pistons in order to extract work independently from the expansion of both gases. The gases can easily be re-separated using this apparatus. In effect, one is then just running two independent heat engines.

In principle, the mixing done by our quantum ignorant observer could also be reversed. However, the non-trivial question is how to perform this as a separate stroke in a heat engine cycle.

Reply to Referee #2

1. *Quantum thermodynamics is the study of the thermodynamic properties of quantum systems. Although the phenomenological framework remains unchanged, surprising and subtle quantum effects may be sought in carefully computing thermodynamic quantities from the underlying quantum states. The present manuscript does just that in the context of the Gibbs paradox. The result is interesting, and the analysis is transparent. The narrative is pleasant to read, and the discussion is convincing. Therefore, I would like to see the manuscript eventually be published in Nature Communications.*

We thank the Referee for their positive comments concerning both our manuscript and its publication in Nature Communications. We have endeavoured to implement the suggestions below in order to further improve the calibre of the manuscript.

2. *Carefully thinking about Gibbs's paradox it seems apparent that the problem of mixing (in-)distinguishable gases is a generalization of Maxwell's demon. Classically, more work can be extracted by the "informed observer", that is by the observer that has taken a measurement with respect to the nature of the gases. The natural question arises how to rephrase the present arguments in terms of proper "quantum observations". In order to distinguish the quantum states (to get the counting right), one has to actually measure the system. The question then is whether one needs to more carefully account for the "cost" of the measurement and the potential quantum back action on the gases states.*

The Referee is correct in thinking that there are parallels between Gibbs mixing and Maxwell's demon. We are indeed studying how thermodynamic work can be extracted based on the knowledge of an observer. However, there are important differences to be clarified.

Crucially, the observer *does not need to make a measurement* during the work extraction protocol. All that the ignorant observer needs to do is engineer an interaction that is dependent on the label J – but they themselves need not acquire any information about it.

The same idea is at work in the classical case when the informed observer extracts work separately from the two gases. They use two spin-dependent semi-permeable pistons to expand the gases independently. This constitutes spin-dependent dynamics, but there is no need to actually measure the spins themselves.

Hence there is no measurement cost associated with the protocol. Of course, the informed observer could measure the spins, or the ignorant observer could measure J if they wanted to (which is connected to their ability to engineer dynamics dependent on these quantities) – but there is no need for this.

This subtle point has been made clearer in the manuscript by altering some of the language away from the term "measurement" and by including a footnote [39]:

Note that the work extraction process does not involve a measurement by the observer – only a coupling to the apparatus that depends on the value of J . Therefore there is no need to consider an additional entropic measurement cost, unlike the case of Maxwell's demon.

3. *The authors assign a spin degree of freedom to the gas particles, which constitutes a label of distinguishability. From a more thermodynamic point of view this spin is nothing else but an information bearing degree of freedom. Thus, a concise discussion in the terminology of (quantum)*

thermodynamics of information might help to make the conceptual part of the analysis a lot more accessible.

We thank the Referee for sign-posting this terminology. Historically we find that such nomenclature was first used by Landauer [16, 17] to discuss a memory register or computer possessing information. It appears that an important aspect of a degree of freedom to qualify as information-bearing is the ability to remain decoupled from the heat bath (at least for a sufficiently long time) to keep its information content stable. In this sense, the spin degree of freedom is indeed being treated as information-bearing.

This provides an insightful way of describing our mixing phenomenon: information is encoded in the initial state in the spins' *relational* degrees of freedom (i.e., whether they are parallel or anti-parallel), and by virtue of (anti-)symmetrisation of the wavefunction, some aspect of this “leaks” into the spatial part, which can be accessed by the ignorant observer.

We have therefore added such terminology and included a reference in the Introduction when discussing the utility of the apparatuses.

One could also describe the spin as an *information-bearing degree of freedom* [16]. The question is whether the information encoded within the spin state has an impact upon the thermodynamics of mixing.

4. The authors introduce the notion of a work battery. Since this is a notion exclusive to the resource theoretic framework to quantum thermodynamics, I wonder why this terminology is used in the present context. The analysis is phrased entirely in a standard language of statistical mechanics, and hence the term work reservoir may be more apt.

We had not meant to imply only a resource-theoretic setting – we had understood “battery” as applying more generally to a system whose energy changes are attributable to work, without entropy changes. This is done in some places in the literature, e.g. [18]. We are however happy to make the change in terminology if the suggested one is more widely used, and have now replaced our previous usage of “battery” with “reservoir” throughout the manuscript.

5. In Eq. (4) the authors find the extractable work as difference of the NONEQUILIBRIUM free energies. It is easy to see that the system is initially prepared in a state of local equilibrium, which relaxes towards a state of global equilibrium. Hence, a more careful formulation of the thermodynamics is crucial. The considered work is actually the excess work, which accounts for the full non equilibrium character of the situation. Also, it may be worthwhile to add a citation for the nonequilibrium free energy, which to the best of my knowledge was introduced in Bergmann & Lebowitz, PR 99, 578 (1955)

The Referee is very much correct that our system begins in two states of local equilibrium which then relax into a state of global equilibrium. Hence, as the system *isn't* moving from one equilibrium state to another the extractable work is not the difference between equilibrium free energies. We find the literature somewhat divided in terminology differentiating (non)equilibrium free energies, some literature explicitly referring to it [7] and some just calling it “free energy” [5], depending on the subfield.

We accept and include the Referee's suggestion and have incorporated both the suggested reference and an additional review article in our introductory section defining the extractable work from our

system. In addition, we have also explicitly stated the local to global equilibrium transformation that the system undergoes:

The gases on either side of the box start in a state of local equilibrium and via mixing approach global equilibrium. We therefore consider the extractable work to be given by the difference in non-equilibrium free energy F [19] between initial and final states...

Reply to Referee #3

1. *These results are both counter-intuitive and interesting, and they add to our understanding of the Gibbs paradox. The analysis in the paper is not straightforward to follow, as it involves somewhat specialized theoretical tools such as Schur-Weyl duality and the Schur basis. Nevertheless, I believe the analysis is correct, and that a careful reader can understand the meaning of the central result of the paper, even if that reader does not follow every step of the analysis. For these reasons I believe this manuscript deserves publication in Nature Communications.*

We thank the Referee for their positive comments regarding our work and the recommendation to publish in Nature Communications. We hope that following our most recent changes to the manuscript, any unclear points previously in the paper have been resolved and the presentation of our results overall has become more transparent for a broad readership.

Collated list of primary changes

Displayed in order of appearance within manuscript.

- Title change:

“Mixing indistinguishable systems leads to a quantum Gibbs paradox”

- Added/reworded text in Results:

This might seem like an unrealistic assumption; however, this model contains the purely combinatorial (or “state-counting”) statistical effects, first analysed by Boltzmann [1], that are known to recover the entropy changes for a classical ideal gas [2–4] using the principle of *equal a priori probabilities*.

- Added text in Results:

Since the particle number is strictly fixed, we are working in the canonical ensemble (rather than the grand canonical ensemble).

- Added text in Results:

The gases on either side of the box start in a state of local equilibrium and via mixing approach global equilibrium. We therefore consider the extractable work to be given by the difference in non-equilibrium free energy F [27]...

- Added text in Results:

The extractable work in a process that takes ρ to ρ' is then

$$W \leq F(\rho) - F(\rho') = k_B T [S(\rho') - S(\rho)]. \quad (2)$$

In a classical reversible process, the extractable work is equal to the change in free energies. This is generally an over-simplification for small systems, in which work can be defined in various ways [7] – e.g. required to be deterministic in the resource theory context [6] or as a fluctuating random variable [8, 9], requiring consideration of other varieties of free energy. However, equation (1) will turn out to be sufficient for our purposes in the sense of mean extractable work. We find the inequality to be saturable using thermal operations and characterise fluctuations around the mean in the latter part of our results section.

- Added text in Results:

One could also describe the spin as an *information-bearing degree of freedom* [16]. The question is whether the information encoded within the spin state has an impact upon the thermodynamics of mixing.

- New table “Summary of the observers’ abilities”

- Added text in Results:

This observer may obtain information about the numbers of spin- \uparrow and spin- \downarrow particles. Thus they can engineer spin-dependent operations conditional on these numbers, but cannot change the number of each spin.

- Added text in Results:

For each J , there exists an operation within the thermal operations framework [6] that performs deterministic work extraction saturating inequality (1). This is because the transformation is between (energy-degenerate) uniformly mixed states of differing dimension.

- Editing of Table 2 “Summary of results”
- New section in Results: “Non-orthogonal spins”
- New Figure 4 plotting “Results for partially distinguishable spins”
- Added text in Discussion:

Two recent papers [12, 13] have studied Gibbs-type mixing in the context of optomechanics. There, a massive oscillator playing the role of a work reservoir interacts with the photons via their pressure. There is also a beam splitter between the two sides of the cavity. In Ref. [12], the beam splitter is non-polarising and thus (together with the interaction with the oscillator) accessible to the ignorant observer. The main behaviour there is driven by the HOM effect, which impedes energy transfer to the oscillator. However, this does not contradict our findings: we have shown that an advantage is gained by optimising over all allowed dynamics. It is therefore an interesting question whether such proposals can be modified to see an advantage of the type described here, even if not optimal. Ref. [13] studies Gibbs mixing as a function of the relative polarisation rotations between left and right. However, this uses a polarising beam splitter, which is only accessible to the informed observer. Therefore the effect described here cannot be seen in such a set-up.

- Added text in Discussion:

It is important to determine how the thermodynamic enhancements predicted in this paper may have implications for physical systems. Such an investigation should make use of more practical proposals (such as Refs. [12–14]) to better understand possible realisations of mixing. For example, systems of ultra-cold atoms in optical lattices [15] may provide a suitable platform to experimentally realise the thermodynamic effects predicted in this work.

- New footnote in References:

Note that the work extraction process does not involve a measurement by the observer - only a coupling to the apparatus that depends on the value of J . Therefore there is no need to consider an additional entropic measurement cost, unlike the case of Maxwell’s demon.

- New Supplementary Note 3 “Partial Distinguishability”

In addition to the above quoted additions, there have been some minor changes to the text for formatting reasons.

-
- [1] L. Boltzmann, Wiener Berichte **76**, 373 (1877).
 [2] O. Darrigol, Entropy **20**, 443 (2018).
 [3] S. Saunders, Entropy **20**, 552 (2018), ISSN 1099-4300, URL <http://www.mdpi.com/1099-4300/20/8/552>.

- [4] D. Dieks, Entropy **20**, 466 (2018), ISSN 1099-4300, URL <http://www.mdpi.com/1099-4300/20/6/466>.
- [5] F. G. Brandao, M. Horodecki, J. Oppenheim, J. M. Renes, and R. W. Spekkens, Physical Review Letters **111**, 250404 (2013).
- [6] M. Horodecki and J. Oppenheim, Nature communications **4**, 2059 (2013).
- [7] W. Niedenzu, M. Huber, and E. Boukobza, Quantum **3**, 195 (2019).
- [8] J. Åberg, Nature Communications **4**, 1 (2013).
- [9] O. C. O. Dahlsten, R. Renner, E. Rieper, and V. Vedral, New Journal of Physics **13**, 053015 (2011), ISSN 1367-2630, URL <https://iopscience.iop.org/article/10.1088/1367-2630/13/5/053015>.
- [10] N. Killoran, M. Cramer, and M. B. Plenio, Physical review letters **112**, 150501 (2014).
- [11] B. Morris, B. Yadin, M. Fadel, T. Zibold, P. Treutlein, and G. Adesso, Phys. Rev. X **10**, 041012 (2020), URL <https://link.aps.org/doi/10.1103/PhysRevX.10.041012>.
- [12] Z. Holmes, J. Anders, and F. Mintert, Physical Review Letters **124**, 210601 (2020).
- [13] Z. Holmes, J. Anders, and F. Mintert, arXiv preprint arXiv:2006.00613 (2020).
- [14] N. M. Myers and S. Deffner, Physical Review E **101**, 012110 (2020), ISSN 2470-0045, URL <https://link.aps.org/doi/10.1103/PhysRevE.101.012110>.
- [15] A. M. Kaufman, M. E. Tai, A. Lukin, M. Rispoli, R. Schittko, P. M. Preiss, and M. Greiner, Science **353**, 794 (2016).
- [16] R. Landauer, IBM journal of research and development **5**, 183 (1961).
- [17] D. Reeb and M. M. Wolf, New Journal of Physics **16**, 103011 (2014).
- [18] F. Campaioli, F. A. Pollock, F. C. Binder, L. Céleri, J. Goold, S. Vinjanampathy, and K. Modi, Physical review letters **118**, 150601 (2017).
- [19] P. G. Bergmann and J. L. Lebowitz, Physical Review **99**, 578 (1955).

Reviewers' Comments:

Reviewer #1:

Remarks to the Author:

The authors have answered my questions and made changes to the paper clarifying the raised issues. The new results on the non-orthogonal spin states are a very nice addition.

In principle the paper can now be accepted for publication.

However, there are a few issues (wrong statements?) affecting the reply and the paper, that the authors may want to check out:

1. In the reply they state "However, what is known from the resource theoretic treatment of thermodynamics is that such a unitary extracting the quoted amount of work definitely exists [5, 6]. This is in the same sense that when one talks about work extraction classically, one often states that there exists an adiabatic transformation that indeed saturates the bound imposed by the 1st law.

In order to make the above clearer we have replaced the equality in equation (1) with an inequality representing the bound imposed by the 1st law and devoted the text preceding and following this equation to how one might optimally saturate this bound both classically and with quantum systems."

Surely they mean: "... one often states that there exists a **reversible** transformation that indeed saturates the bound imposed by the **2nd law**."

In order to make the above clearer we have replaced the equality in equation (1) with an inequality representing the bound imposed by the **2nd law** ... " ???

2. In the reply/paper they state: "Two recent papers [12, 13] have studied Gibbs-type mixing in the context of optomechanics. There, a massive oscillator playing the role of a work reservoir interacts with the photons via their pressure. There is also a beam splitter between the two sides of the cavity. In Ref. [12], the beam splitter is non-polarising and thus (together with the interaction with the oscillator) accessible to the ignorant observer. The main behaviour there is driven by the HOM effect, which impedes energy transfer to the oscillator. However, this does not contradict our findings: we have shown that an advantage is gained by optimising over all allowed dynamics."

Actually, in paper [12] the oscillator is a membrane which **IS** the beamsplitter, and the paper reports that the HOM **enhances** (not impedes) energy transfer.

Ref[12]=Enhanced energy transfer to an optomechanical piston from indistinguishable photons, Z Holmes et al, PRL 2020.

Point 3 had nothing to do with grand canonical.

Reviewer #2:

Remarks to the Author:

I already mentioned in my previous report that I would like to see the manuscript published in Nature Communications. In their revisions that authors carefully and thoroughly addressed all issues raised by the referees. Therefore, I can now recommend to accept the current version for publication.

Signed: Sebastian Deffner

Reply to Referee #1

1. *In the reply they state “However, what is known from the resource theoretic treatment of thermodynamics is that such a unitary extracting the quoted amount of work definitely exists [5, 6]. This is in the same sense that when one talks about work extraction classically, one often states that there exists an adiabatic transformation that indeed saturates the bound imposed by the 1st law. In order to make the above clearer we have replaced the equality in equation (1) with an inequality representing the bound imposed by the 1st law and devoted the text preceding and following this equation to how one might optimally saturate this bound both classically and with quantum systems.”*

Surely they mean: “... one often states that there exists a reversible transformation that indeed saturates the bound imposed by the 2nd law. In order to make the above clearer we have replaced the equality in equation (1) with an inequality representing the bound imposed by the 2nd law...” ???

The referee is of course correct, we meant the 2nd law. We apologise for this unfortunate typo in the reply. The same also goes for “adiabatic”, which has been replaced in the main text by referring only to free operations in the resource theory.

2. *In the reply/paper they state: “Two recent papers [12, 13] have studied Gibbs-type mixing in the context of optomechanics. There, a massive oscillator playing the role of a work reservoir interacts with the photons via their pressure. There is also a beam splitter between the two sides of the cavity. In Ref. [12], the beam splitter is non-polarising and thus (together with the interaction with the oscillator) accessible to the ignorant observer. The main behaviour there is driven by the HOM effect, which impedes energy transfer to the oscillator. However, this does not contradict our findings: we have shown that an advantage is gained by optimising over all allowed dynamics.”*

Actually, in paper [12] the oscillator is a membrane which IS the beamsplitter, and the paper reports that the HOM enhances (not impedes) energy transfer.

We thank you for clarifying the approach in [1]. We had mixed up aspects of the two different references, and wished instead to refer to the fact that in [2] it is stated that “the effects of bunching and Gibbs mixing are antithetical” – which is relevant because this appears to be different to what is found in our scheme. We have therefore amended our comments in this section to read:

Two recent papers [1, 2] have studied Gibbs-type mixing in the context of optomechanics. A massive oscillator playing the role of a work reservoir interacts with the photons via their pressure. This oscillator simultaneously acts as a beam splitter between the two sides of the cavity. In Ref. [1], the beam splitter is non-polarising and thus (together with the interaction with the oscillator) accessible to the ignorant observer. The main behaviour there is driven by the HOM effect, which enhances the energy transfer to the oscillator, albeit in the form of fluctuations. In Ref. [2], which studies Gibbs mixing as a function of the relative polarisation rotations between left and right, bosonic statistics are therefore described as acting oppositely to Gibbs mixing effects – which is different from our conclusions. However, there is no contradiction: we have shown that an advantage is gained by optimising over all allowed dynamics. Moreover, the scheme in Ref. [2] uses a polarisation-dependent beam splitter, which is only accessible to the informed observer. Therefore the effect described here cannot be seen in such a set-up. It is an interesting question whether such proposals can be modified to see an advantage of the type described here, even if not optimal.

[1] Z. Holmes, J. Anders, and F. Mintert, Physical Review Letters **124**, 210601 (2020).

[2] Z. Holmes, F. Mintert, and J. Anders, New Journal of Physics **22**, 113015 (2020), URL <https://doi.org/10.1088/1367-2630/abc602>.